# Computed tomography-derived radiomic signature of head and neck squamous cell carcinoma (peri)tumoral tissue for the prediction of locoregional recurrence and distant metastasis after concurrent chemo-radiotherapy

Simon Keek[1], Sebastian Sanduleanu[1]*, Frederik Wesseling[2], Reinout de Roest[3], Michiel van den Brekel[4,5], Martijn van der Heijden[4,6], Conchita Vens[6,7], Calareso Giuseppina[8], Lisa Licitra[9,10], Kathrin Scheckenbach[11], Marije Vergeer[12], C. René Leemans[3], Ruud H Brakenhoff[3], Irene Nauta[3], Stefano Cavalieri[8], Henry C. Woodruff[1,13], Tito Poli[14], Ralph Leijenaar[2], Frank Hoebers[2], Philippe Lambin[1,13]

1 The D-lab, Department of Precision Medicine, GROW – School for Oncology and Developmental Biology, Maastricht University, Maastricht, The Netherlands, 2 Department of Radiation Oncology (MAASTRO), GROW – School for Oncology and Developmental Biology, Maastricht University Medical Centre+, Maastricht, The Netherlands, 3 Amsterdam UMC, Vrije Universiteit Amsterdam, Otolaryngology / Head and Neck Surgery, Cancer Center Amsterdam, The Netherlands, 4 Department of Head and Neck Oncology and Surgery, The Netherlands Cancer Institute, Amsterdam, The Netherlands, 5 Department of Oral and Maxillofacial Surgery, Academic Medical Center, Amsterdam, The Netherlands, 6 Division of Cell Biology, The Netherlands Cancer Institute, Amsterdam, The Netherlands, 7 Department of radiation oncology, The Netherlands Cancer Institute, Amsterdam, The Netherlands, 8 Istituto nazionale dei tumori, Department of Radiology, Milan, Italy, 9 Fondazione IRCCS Istituto Nazionale dei Tumori, Head and Neck Medical Oncology Department, Milan, Italy, 10 University of Milan, Department of Oncology and Hematology-Oncology, Milan, Italy, 11 Dept. of Otorhinolaryngology, Head and Neck Surgery, Heinrich-Heine-University, Düsseldorf, Germany, 12 Amsterdam UMC, Vrije Universiteit Amsterdam, Department of Radiation Oncology Amsterdam, The Netherlands, 13 Department of Radiology and Nuclear Medicine, GROW – School for Oncology and Developmental Biology, Maastricht University Medical Centre+, Maastricht, The Netherlands, 14 University of Parma, Department of Surgical Sciences, Parma, Italy

☯ These authors contributed equally to this work.
* s.sanduleanu@maastrichtuniversity.nl

## Abstract

### Introduction

In this study, we investigate the role of radiomics for prediction of overall survival (OS), loco-regional recurrence (LRR) and distant metastases (DM) in stage III and IV HNSCC patients treated by chemoradiotherapy. We hypothesize that radiomic analysis of (peri-)tumoral tissue may detect invasion of surrounding tissues indicating a higher chance of locoregional recurrence and distant metastasis.

### Methods

Two comprehensive data sources were used: the Dutch Cancer Society Database (Alp 7072, DESIGN) and "Big Data To Decide" (BD2Decide). The gross tumor volumes (GTV)

**Data Availability Statement:** Data cannot be shared publicly because we are not allowed to share the imaging data as third party (2 year moratorium on BD2DECIDE data). Radiomics/ clinical data and script are available from Github https://github.com/SebastianSanduleanu/ Peritumoral-HN-Radiomics.git for researchers who meet the criteria for access to confidential data.

**Funding:** Funded: DESIGN: Alpe d'Huzes/ KWF Program Grant A6C 7072. BD2DECIDE: European Union Horizon 2020 research/innovation program (689715). Dr. R Leijenaar received support in the form of a salary from OncoRadiomics. The specific roles of these authors are articulated in the 'author contributions' section.

**Competing interests:** Dr. Philippe Lambin reports, within and outside the submitted work, grants/ sponsored research agreements from Varian medical, Oncoradiomics, ptTheragnostic, Health Innovation Ventures and DualTpharma. He received an advisor/presenter fee and/or reimbursement of travel costs /external grant writing fee and/or in kind manpower contribution from Oncoradiomics, BHV, Merck and Convert pharmaceuticals. Dr Lambin has shares in the company Oncoradiomics SA and Convert pharmaceuticals SA and is coinventor of two issued patents with royalties on radiomics (PCT/ NL2014/050248, PCT/NL2014/050728) licensed to Oncoradiomics and one issue patent on mtDNA (PCT/EP2014/059089) licensed to ptTheragnostic/ DNAmito, three nonpatentable invention (softwares) licensed to ptTheragnostic/DNAmito, Oncoradiomics and Health Innovation Ventures. Dr. Woodruff has (minority) shares in the company Oncoradiomics. CR Leemans and RH Brakenhoff have received financial support from GenMab BV, InteRNA Technologies BV and Bristol Myers -Squibb, all outside the presented work. Dr. R Leijenaar has OncoRadiomics shares and is an employee of the company. This does not alter our adherence to PLOS ONE policies on sharing data and materials.

were delineated on contrast-enhanced CT. Radiomic features were extracted using the RadiomiX Discovery Toolbox (OncoRadiomics, Liege, Belgium). Clinical patient features such as age, gender, performance status etc. were collected. Two machine learning methods were chosen for their ability to handle censored data: Cox proportional hazards regression and random survival forest (RSF). Multivariable clinical and radiomic Cox/ RSF models were generated based on significance in univariable cox regression/ RSF analyses on the held out data in the training dataset. Features were selected according to a decreasing hazard ratio for Cox and relative importance for RSF.

## Results

A total of 444 patients with radiotherapy planning CT-scans were included in this study: 301 head and neck squamous cell carcinoma (HNSCC) patients in the training cohort (DESIGN) and 143 patients in the validation cohort (BD2DECIDE). We found that the highest performing model was a clinical model that was able to predict distant metastasis in oropharyngeal cancer cases with an external validation C-index of 0.74 and 0.65 with the RSF and Cox models respectively. Peritumoral radiomics based prediction models performed poorly in the external validation, with C-index values ranging from 0.32 to 0.61 utilizing both feature selection and model generation methods.

## Conclusion

Our results suggest that radiomic features from the peritumoral regions are not useful for the prediction of time to OS, LR and DM.

## Introduction

Head and neck squamous cell carcinoma (HNSCC) is the sixth most common malignant disease worldwide [1]. In the Netherlands, approximately 39,000 men and women were diagnosed with HNSCC between 2000 and 2015 [2]. Roughly two thirds of patients have advanced stage of disease at diagnosis with debilitating symptoms.

Major progress has been made in the treatment of advanced HNSCC throughout the last decade [6]. The "traditional" treatment of these advanced tumors consists of surgical excision followed by complementary (adjuvant) radiotherapy or chemoradiotherapy (CRT). CRT either applied upfront or postoperatively significantly improves survival in HNSCC patients with overall 5-year survival rates up to 61%, 41%, and 69% for oral, pharyngeal and laryngeal cancers, respectively [3–6]. The introduction of organ-preserving therapies (induction chemotherapy, upfront concomitant CRT, or molecular targeted drugs such as cetuximab) has notably changed treatment protocols of advanced stage HNSCC patients, especially in patients where surgical resection is considered too invasive and where severe problems with speech and swallowing are expected after surgery. Concomitant CRT consists of systemic administration of cisplatin in combination with locoregional radiotherapy and is the mainstay of organ-preserving treatment for advanced HNSCC.

It has been shown that 40% of patients treated upfront with CRT develop a locoregional recurrence or distant metastasis within 2 years after treatment and consequently have an unfavorable prognosis [7]. Several studies have found that advanced and human papillomavirus (HPV)-16-negative tumors respond poorly to CRT in contrast to HPV positive tumors, in

particular in oropharyngeal HNSCC [4, 8]. TNM classifications are expected to support patient prognosis by clinicians but unfortunately, they are not helpful to accurately predict which HNSCC patients treated with CRT will develop locoregional recurrences and hence might have benefited from alternative treatment options. Several other potentially prognostic factors have been proposed, such as chemotherapy dose, radiotherapy dose, co-morbidity, World Health Organization (WHO) Performance Status (PS), and HPV-status. Through the use of machine learning algorithms, complex survival models can be created that take these clinical factors into account, while accounting for e.g. interaction between the predictors and right censored data [9].

Currently used biomarkers comprise tumor size, local tumor extent and a few molecular markers (e.g. p16 staining or HPV-PCR). Radiologic imaging, which is routinely performed prior to initiation of CRT, provides an additional source of information that can be exploited through the use of advanced image analysis methods such as radiomics. Radiomics turns radiographic images into a high-throughput data-mining format. The format of the extracted data is a set of features, including first-order intensity histogram statistics, shape- and size statistics, and (filtered) texture features. Complex models that combine radiomics with clinical parameters may be better in detecting HNSCC patients that have a higher likelihood to relapse early after CRT [10].

A growing body of research shows that the tumor microenvironment is a key player in head and neck cancer development and progression [11,12] and hence the immediate surroundings of the tumor may be a source for the extraction of imaging biomarkers. One of the hypotheses is that information about underlying malignancy-associated changes (MAC's) in the tumor microenvironment can be detected by these imaging biomarkers. These MAC changes are subtle changes in the nuclear morphology and chromatin structure of seemingly normal cells located within the stroma distally to neoplastic lesions that have been shown to dictate its ability to grow and spread, evade the body's immune defenses, and resist therapeutic intervention [13].

In this study, we aim to investigate the role of radiomics for prediction of overall survival (OS), locoregional recurrence (LRR) and distant metastasis (DM) in stage III and IV HNSCC patients, both in a HPV-negative oropharyngeal cohort (high risk) as well as in the general HNSCC population. We hypothesize that radiomic analysis of peritumoral tissue detects changes associated with malignancy and therefore the likelihood of locoregional recurrence and distant metastasis following CRT.

## Methods

### Patient characteristics

Two sources of clinical and imaging data were available to us for this study: the Dutch Cancer Society Database (Alp 7072, acronym DESIGN) and "Big Data To Decide" (BD2Decide, NCT02832102). DESIGN is a Dutch multi-center clinical study to create predictive models for stage III and IV HPV-negative HNSCC patients treated by CRT. BD2Decide is a European multi-center clinical study to improve clinical decision making in stage III and IV HNSCC patients irrespective of treatment. In the present study, we included patients from both consortiums with pathologically-confirmed HNSCC, who received contrast-enhanced pre-treatment CT and have been treated upfront with CRT.

The DESIGN data consists of contrast enhanced CT images (and associated clinical data) acquired from 4 different centers: Amsterdam UMC location VUmc, Netherlands Cancer Institute (NKI), Maastricht Radiation Oncology Clinic (MAASTRO), and the University Medical Center Utrecht (UMCU). The BD2Decide data consists of contrast-enhanced CT images

retrospectively acquired from 4 different centers: Fondazione IRCCS Istituto dei Tumori Milano (INT), Maastricht Radiation Oncology Clinic (MAASTRO), Amsterdam UMC, location VUmc (VUMC), and the Heinrich-Heine-university in Düsseldorf. There were no overlapping patients between DESIGN and BD2DECIDE.

Both DESIGN and BD2Decide data included clinical, pathological, radiologic imaging, and molecular markers for each case. After comparing datasets, a selection was made to include patients based on the overlap of available clinical data between the two cohorts. These consist of age, sex, performance status, ACE-27 baseline comorbidity, number of pack years, alcohol consumption, hemoglobin at baseline, chemotherapy regimen, HPV status (defined as p16-status) for oropharyngeal cancer, induction chemotherapy (yes/ no), chemotherapy completion (yes/no), and RT dose to the high-risk clinical target volume (HR-CTV).

## CT acquisition parameters and segmentation

Patients were selected according to the following inclusion criteria: (i) concomitant CRT of unresected HNSCC, (ii) hypopharyngeal, laryngeal or (HPV-negative on p16 staining) oropharyngeal, (iii) no prior treatment with chemotherapy or with radiotherapy in the head and neck area, (iv) availability of contrast-enhanced baseline planning CT imaging with a slice thickness ≤ 5mm and artifacts in less than 50% of the GTV slices, and (v) availability of patient outcome data for OS, LRR, and DM. A large selection of different scanners were used to acquire the images (S1 Appendix).

GTVs were delineated in each center by an assigned radiation oncologist or radiologist. All contours were revised by a radiation oncologist with over 18 years experience, using MIM software version 6.9.0 (MIM, Cleveland, United States).

Tumor border regions of interest (ROI) extending 3mm and 5mm from the 3D GTV border were generated in MIM (outer ring expansion, see Fig 1). Afterwards, air and bone were filtered from the delineation by setting minimum and maximum thresholds, and manually adjusting the final ROI's border (peritumoral) regions.

## Ethical approval

This study was performed following the guidelines of the Code of Conduct for Human Tissue and Medical Research (https://www.federa.org/codes-conduct) and the EU General Data Protection Regulation.

Medical Ethics Committee approval was provided by the individual centers (full list provided in S2 Appendix).

Written informed consent was given and was placed under the responsibility of the Principal Investigators of the relevant Clinical Participating Centers mentioned above and remain under the custodianship of the specific Participating Centers.

For reproducibility purposes, our code can be found on: https://github.com/PeritumoralRadiomics/Peritumoral-radiomics-HN.git.

## Clinical outcome

The clinical endpoints evaluated in this study were overall survival (OS), locoregional recurrence (LRR) and distant metastasis (DM). The missForest (non-parametric missing value imputation using Random Forest) function within the R environment (https://www.R-project.org/) was used to impute missing data. Time to OS was defined as the time between CRT start date and date of death, or censored at the last follow-up date.

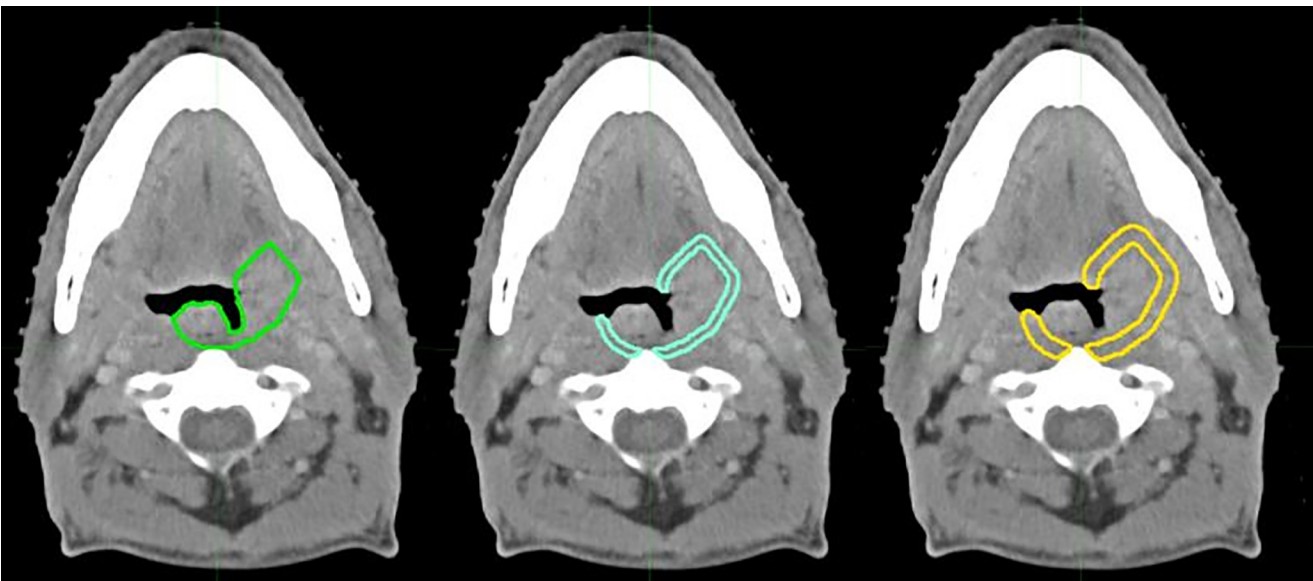

**Fig 1. Contrast-enhanced CT image from an oropharyngeal cancer patient.** Primary gross tumor volume (GTV1) border in green, blue: 3mm peritumoral border, yellow: 5mm peritumoral border.

Time to LRR was defined as the time between CRT start date and the first scan date of radiologically evident local or regional recurrence (event), or censored at the last follow-up date or date of death.

Time to DM was defined as the time between CRT start date and the first scan date of radiologically evident distant metastasis, or censored at the last follow-up date or date of death.

## Image pre-processing, radiomic feature extraction and feature harmonization

International Biomarker Standardization Initiative (IBSI)-compliant radiomic features as well as other non-IBSI covered features were extracted with our in-house RadiomiX research software (supported by Oncoradiomics, Liège, Belgium) implemented in Matlab 2017a (Mathworks, Natick, Mass). Hounsfield Unit (HU) intensities beyond -1024 and +3071 HU were clipped (assigned the value -1024 and +3071 respectively). An image intensity discretization applying a fixed bin width of 25HU was used for feature extraction in CT. Voxel size resampling was performed before feature extraction using cubic interpolation. Images were resampled to isotropic voxels of size 3 x 3 x 3 mm$^3$ using cubic interpolation (upsampling to highest slice thickness).

Radiomic features were extracted consisting of five main groups: 1) fractal features 2) first order statistics, 3) shape and size, 4) texture descriptors including gray level co-occurrence (GLCM), gray level run-length (GLRLM) and gray level size-zone texture matrices (GLSZM), 5) features from groups 1, 3 and 4 after wavelet decomposition of the original image. There were no missing feature values. Definitions and detailed feature descriptions are described elsewhere [14].

Radiomic feature values are potentially sensitive to inter-scanner model, acquisition protocol and reconstruction settings variations. The ComBat statistical feature harmonization technique was employed in our analysis. This technique was initially developed by Johnson et al. [15] for gene expression microarray data (even for small sample sizes) and was recently applied

in multicenter PET, MRI and CT radiomic studies [16,17]. Feature values were adjusted for the batch effect according to treatment center, without adjustment for other covariates. Finally, features were normalized in the training dataset by the mean and standard deviation, which were subsequently used to normalize the validation dataset.

### Univariable analysis and generation of multivariable models

The prognostic value of the individual radiomic and clinical features was evaluated using concordance index (CI) with the survival package (Therneau T (2015). A Package for Survival Analysis in R. version 2.38, URL: https://CRAN.R-project.org/package=survival) and random-ForestSRC package (Ishwaran H (2017) Fast Unified Random Forests for Survival, Regression and Classification (RF-SRC) version 2.9.1, URL: https://cran.r-project.org/web/packages/randomForestSR).

Noether's method was applied to assess the statistical significance of the computed CI from random chance (CI = 0.5) with the survcomp package (Benjamin Haibe-Kains (2017). Performance Assessment and Comparison for Survival Analysis in R. version 1.36.0, URL: https://www.pmgenomics.ca/bhklab/). To account for multiple testing, a false-discovery-rate (FDR) procedure by Benjamin and Hochberg was applied to adjust the p-values in univariate Cox-regression.

Two machine learning methods were employed that are able to use censored survival data as inputs: Cox proportional hazards based and random survival forrest (RSF).

Multivariable radiomic Cox models were generated using the significant features selected through univariate cox modelling on the training dataset. In a 100-repeat 2-fold cross-validation on the training data, significant features were selected based on univariate significance (p<0.05) adjusted for multiple testing.

"These features were then ranked according to adjusted hazard ratios, where hazard ratios lower than 1 were inversed, and were gradually added to a multivariate cox model until the first peak in the cross-validation testing C-index or after the first peak until the C-index drops by more than 0.02, depending if there is an oscillation or noise pattern leading to multiple peaks. The number of occurrences of each feature in all repetitions was determined, and a selection rate > 50% was used as threshold for the final set of features, ensuring that the selected features were chosen in the majority of the models."

Multivariable clinical models included features selected through Cox-regression based on univariate significance (p<0.05) adjusted for multiple testing. The selected clinical features were then used to train multivariable Cox or RSF models.

Multivariable clinical RSF models were generated based on selecting all features with a relative feature importance >0 in the Random Survival Forest.

RSF strictly adheres to the prescription laid out by Breiman (2003) and requires taking into account the outcome (splitting criterion used in growing a tree must explicitly involve survival time and censoring information) in growing a random forest model. Further, the predicted value for a terminal node in a tree, the resulting ensemble predicted value from the forest, and the measure of prediction accuracy must all properly incorporate survival information.

Multivariable radiomic RSF models were generated based on the optimal number of features corresponding to the first peak in C-index value in the out-of-bag cases OR after the first peak until the C-index frops by more than 0.02, depending if there is an oscillation or noise pattern leading to multiple peaks. Hereby features with decreasing relative importance in the Random Survival Forest were consecutively added.

## Results

### Clinical characteristics

Contrast enhanced CT images from a total of 444 patients were included in this study: The training cohort (DESIGN) consisted of 301 head and neck squamous cell carcinoma (HNSCC) patients and the validation cohort (BD2DECIDE) of 143 patients. At time of diagnosis, the median age in the training cohort (DESIGN) was 61 years (range: 36 to 80 years), while the median age in the external validation cohort (BD2DECIDE) was 60.5 years (range: 41 to 78 years).

In the training dataset the median OS time was 1118 days, the median time to LRR or last follow-up was 1042 days and the median time to DM or last follow-up was 1060 days. In the external validation dataset the median time to death or last follow-up was 1268 days, the median time to LRR or last follow-up was 1217 days and the median time to DM or last follow-up was 1189 days.

The full list of patient characteristics and time to progression is presented in Table 1.

### Clinical characteristics

Clinical models (Tables 2 and 3) to predict OS, LR and DM ranged from a C-index of 0.61–0.85 in training with both methods and a C-index of 0.49–0.75 in external validation. Details on the clinical variable selected in the final Cox/ RSF models are presented in Table 4.

The highest performing model in external validation was a clinical model (Oropharynx-DM). With this clinical model a significant survival split was found both in training (Fig 2a) but not in validation (Fig 2b) based on the median prediction probabilities in training according to the Cox model.

### Radiomics characteristics

A total of 1298 radiomic features were extracted from all contrast-enhanced CT-images. Results of training (DESIGN) and validation (BD2DECIDE) c-index metrics are provided in Tables 2 and 3. Both in oropharyngeal cases alone as well as in all tumor subsites combined peritumoral radiomics performed poorly in external validation, with C-index ranging from 0.32 to 0.61 with both feature selection and model generation methods. (Figs 3 and 4).

Volumetric information was calculated for $GTV_{prim}$ and Spearman correlation coefficients between individual selected features and volume were calculated. With the Cox method these C-indexes were all <0.60 (all P>0.05 correlation with model features). With the RSF method these varied between 0.28–0.45 (all P>0.05 correlation with model features).

### Radiomics quality assurance and TRIPOD statement

For quality assurance a radiomics quality score (RQS) was calculated [14] for this study. The RQS score for this specific study was 44% (most points allocated for external validation and use of feature reduction analysis).

Scores were likewise calculated for the 22-item adherence data extraction checklist of the TRIPOD (Transparent reporting of a multivariable prediction model for individual prognosis or diagnosis), which was in the range of 0.75–0.86 (See S3 Appendix).

## Discussion

In this first peritumoral H&N radiomics study we found that the highest performing model in external validation was a clinical model which was able to predict distant metastasis in oropharyngeal cancer cases with an external validation c-index of 0.65 and 0.75 with the RSF and Cox

**Table 1. DESIGN/ BD2DECIDE patient characteristics.**

| | DESIGN training cohort (n = 301) | BD2DECIDE validation cohort (n = 143) | P-value |
|---|---|---|---|
| | Median (range) | Median (range) | |
| GTV$_{prim}$ Volume (cm$^3$) | 21.28 (0.65–176.10) | 19.82 (0.54–157,28) | 0.82 |
| **Age (years)** | 61 (36–80) | 60 (41–78) | 0.52 |
| | Number of pts (%) | Number of pts (%) | |
| **WHO PS** | | | <0.001 |
| 0 | 0 (0) | 120 (83.9) | |
| 1 | 79 (26.2) | 20 (14.0) | |
| 2 | 139 (46.2) | 3 (2.1) | |
| 3 | 10 (3.3) | 0 (0) | |
| Missing | 73 (24.3) | 0 (0) | |
| **Clinical TNM (T), 7th Edition** | | | 0.08 |
| cTX | 0 (0) | 0 (0) | |
| cT1 | 14 (4.7) | 3 (2.1) | |
| cT2 | 63 (20.9) | 25 (17.5) | |
| cT3 | 106 (35.2) | 68 (47.6) | |
| cT4 | 118 (39.2) | 47 (32.9) | |
| **Clinical Nodal stage (N), 7th Edition** | | | 0.01 |
| cNX | 1 (0.3) | 0 (0) | |
| cN0 | 41 (13.6) | 37 (25.9) | |
| cN1 | 41 (13.6) | 19 (13.3) | |
| cN2 a-b-c | 209 (69.5) | 79 (55.2) | |
| cN3 | 9 (3.0) | 8 (5.6) | |
| **HPV status (P16 stain)** | | | <0.001 |
| Negative | 207 (68.8) | 64 (44.8) | |
| Positive/ Unknown | 94 (31.2) | 79 (55.2) | |
| **Treatment** | | | |
| **Chemotherapy regimen** | | | <0.001 |
| **• Platin** | 292 (97.0) | 81 (56.6) | |
| **• Platin + others** | 9 (3.0) | 23 (16.1) | |
| **• Cetuximab** | 0 (0) | 39 (27.3) | |
| **Cumulative radiotherapy dose high-risk CTV** | 70 (60–84) Gy | 70 (20–76) Gy | |
| **Tumor site** | | | |
| Oropharynx | 145 (48.2) | 49 (34.3) | 0.02 |
| Larynx | 57 (18.9) | 39 (27.3) | |
| Hypopharynx | 99 (32.9) | 55 (38.5) | |

models respectively. Both in oropharyngeal cases alone as well as in all tumor subsites peritumoral radiomics performed poorly in external validation, with C-index ranging from 0.32 to 0.61 with both feature selection and model generation methods.

The reasoning for choosing a 5mm tumor border is based on radiotherapy margins which are defined outside the visible/palpable or imaging-detectable (macroscopic) tumor GTV, the clinical target volume (CTV), whereby potential microscopic tumor spread is taken into account. Based on experience from pathological examination of surgical resections, the Danish Head and Neck Cancer (DAHANCA) group concluded that for primary tumors (GTV-T), the risk of subclinical microscopic spread was around 50% of which more than 99% was within 5 mm and 95% within 4 mm of the rim of GTV-T [18].

**Table 2. Multivariable Cox Regression method, C-index and number of radiomic and (non)-treatment related prognostic clinical factors in validation dataset (BD2DECIDE).**

| | C-index Prognostic (No. feat) | | C-index GTV$_{prim}$ (No. feat) | | C-index TB 3mm (No. feat) | | C-index TB 5mm (No. feat) | | C-index GTV$_{prim}$, + TB 3mm + TB 5mm (No. feat) | |
|---|---|---|---|---|---|---|---|---|---|---|
| | Train | Val | Train | Val | Train | Val | Train | Val | Train | Val |
| **Oropharynx** | | | | | | | | | | |
| Clinical-OS | 0.61 (1) | 0.49 (1) | | | | | | | | |
| Clinical-LR | 0.61 (1) | 0.55 (1) | | | | | | | | |
| Clinical-DM | 0.67 (1) | 0.65 (1) | | | | | | | | |
| Radiomics-OS | | | 0.65 (3) | 0.57 (3) | 0.69 (3) | 0.52 (3) | 0.79 (1) | 0.60 (1) | 0.70 (2) | 0.56 (2) |
| Radiomics-LR | | | 0.57 (1) | 0.52 (1) | 0.70 (2) | 0.56 (2) | 0.76 (6) | 0.51(6) | 0.72 (4) | 0.48 (4) |
| Radiomics-DM | | | - | - | 0.69 (2) | 0.61 (2) | 0.73 (3) | 0.44 (3) | 0.72 (2) | 0.60 (2) |
| **All subsites** | | | | | | | | | | |
| Clinical-OS | 0.64 (4) | 0.56 (4) | | | | | | | | |
| Clinical-LR | - | - | | | | | | | | |
| Clinical-DM | 0.67 (1) | 0.49 (1) | | | | | | | | |
| Radiomics-OS | | | 0.61 (1) | 0.60 (1) | 0.63 (4) | 0.61 (4) | 0.61 (2) | 0.62 (2) | 0.61 (3) | 0.59 (3) |
| Radiomics-LR | | | 0.66 (3) | 0.51 (3) | 0.67 (3) | 0.51 (3) | 0.58 (1) | 0.47 (1) | 0.61 (1) | 0.47 (1) |
| Radiomics-DM | | | 0.63 (2) | 0.54 (2) | 0.54 (2) | 0.47 (4) | 0.61 (2) | 0.56 (2) | 0.64 (3) | 0.55(2) |

Abbreviations GTV$_{prim}$—Primary Gross Tumor Volume, OS- Overall Survival, LR- Locoregional Recurrence, DM- Distant Metastasis.

Previous studies on peritumoral radiomics in other tumor models have not been able to produce promising results in internal cross-validation either. We have not yet seen a peritumoral H&N radiomics study with an external validation dataset.

Dou et al. [19] for instance found a testing C-index of 0.55 with a lung radiomic tumor border model in the prediction of distant metastasis, while Shan et al. [20] found that in predicting

**Table 3. Random survival forest method, C-index and number of radiomic or (non)-treatment related prognostic clinical factors.**

| | C-index Prognostic (No. feat) | | C-index Treatment (No. feat) | | C-index GTV$_{prim}$ (No. feat) | | C-index TB 3mm (No. feat) | | C-index TB 5mm (No. feat) | | C-index GTV$_{prim}$, + TB 3mm + TB 5mm (No. feat) | |
|---|---|---|---|---|---|---|---|---|---|---|---|---|
| | Train | Val | Train | Val | Train | Val | Train | Val | Train | Val | Train | Val |
| **Oropharynx** | | | | | | | | | | | | |
| Clinical-OS | 0.74 (5) | 0.74 (5) | 0.52 (1) | 0.53 (1) | | | | | | | | |
| Clinical-LR | 0.81 (5) | 0.81 (5) | 0.51 (1) | 0.51 (1) | | | | | | | | |
| Clinical-DM | 0.85 (4) | 0.85 (4) | 0.51 (1) | 0.52 (1) | | | | | | | | |
| Radiomics-OS | | | | | 0.73 (3) | 0.58 (3) | 0.77 (6) | 0.49 (6) | 0.79 (5) | 0.60 (5) | 0.78 (6) | 0.61 (6) |
| Radiomics-LR | | | | | 0.77 (2) | 0.49 (2) | 0.83 (3) | 0.43 (3) | 0.83 (2) | 0.59 (7) | 0.71 (3) | 0.57 (3) |
| Radiomics-DM | | | | | 0.82 (2) | 0.49 (2) | 0.91 (8) | 0.55 (8) | 0.81 (3) | 0.50 (3) | 0.86 (4) | 0.32 (4) |
| **All subsites** | | | | | | | | | | | | |
| Clinical-OS | 0.77 (7) | 0.77 (7) | 0.56 (1) | 0.51 (1) | | | | | | | | |
| Clinical-LR | 0.79 (3) | 0.79 (3) | 0.56 (2) | 0.49 (2) | | | | | | | | |
| Clinical-DM | 0.84 (4) | 0.84 (4) | - | - | | | | | | | | |
| Radiomics-OS | | | | | 0.79 (7) | 0.58 (4) | 0.89 (4) | 0.58 (4) | 0.77 (5) | 0.60 (5) | 0.78 (6) | 0.59 (6) |
| Radiomics-LR | | | | | 0.81 (3) | 0.52 (2) | 0.80 (2) | 0.52 (2) | 0.86 (7) | 0.59 (7) | 0.83 (2) | 0.53 (2) |
| Radiomics-DM | | | | | 0.86 (3) | 0.49 (3) | 0.86 (4) | 0.49 (4) | 0.96 (3) | 0.50 (3) | 0.86 (3) | 0.43 (3) |

Abbreviations GTV$_{prim}$—Primary Gross Tumor Volume, OS- Overall Survival, LR- Locoregional Recurrence, DM- Distant Metastasis.

**Table 4. Multivariable clinical Cox/ RSF models.**

| Outcome | Clinical Cox, all subsites *Prognostic* | Clinical Cox, Oropharynx *Prognostic* | Clinical RSF, all subsites *Prognostic* | Clinical RSF, Oropharynx *Prognostic* | Clinical RSF, all subsites *Treatment* | Clinical RSF, Oropharynx *Treatment* |
|---|---|---|---|---|---|---|
| **OS** | N-stage | N-stage | N-stage | N-stage | Chemotherapy regimen | Chemotherapy regimen |
| | Tumor site | | Tumor site | Age | Chemotherapy completion | |
| | Gender | | Hb baseline | Pack Years | | |
| | Alcohol consumption | | Age | Alcohol consumption | | |
| | | | Pack-years | Gender | | |
| **LR** | N-stage | Gender | Hb baseline | Gender | Chemotherapy regimen | Chemotherapy regimen |
| | | | Tumor site | Alcohol consumption | Chemotherapy completion | |
| | | | Gender | Age | | |
| | | | | Pack years | | |
| | | | | N-stage | | |
| **DM** | N-stage | N-stage | N-stage | N-stage | | Chemotherapy regimen |
| | | | T-stage | T-stage | | |
| | | | Hb baseline | Age | | |
| | | | Pack-years | Pack years | | |

early recurrence in hepatocellular carcinoma (HCC), by comparing AUC values between training and validation cohorts, the prediction accuracy in the validation cohort was good for the peritumoral radiomics model (0.80 vs. 0.79, P = 0.47) but poor for the tumoral radiomics model (0.82 vs. 0.62, P < 0.01).

Despite the poor performance in external validation with both $GTV_{prim}$, 3mm, and 5 mm tumor border radiomics, we have found a clinical model for the prediction of distant metastasis in oropharyngeal cancer patients performed the best in external validation.

We find an overlapping clinical parameter, namely node-stage, between these two clinical models. Indeed high node stage is hypothesized to be one of the major risk factors for the development of distant metastasis [21,22]. We also see some discrepancies between the two clinical models. For instance, T-stage, age, and packyears (the number of packs of cigarettes per day multiplied by the years spent smoking) are also selected as one of the predictors of distant metastasis in the RSF model.

Strengths of the current study include the use of an external validation dataset, the extensive clinical data and the rigorous feature selection methods that take into account time-to-event outcomes.

One of the limitations is the retrospective nature of the study, leading to several clinical variables (e.g. weight loss) to not be comparable between training and validation. Another limitation is the heterogeneity between the training and validation dataset, both in terms of WHO PS, N-stage, chemotherapy regimen (mostly platin alone regimens in DESIGN versus platin + other regimens in BD2DECIDE) as well as tumor site (DESIGN more oropharynx, less laryngeal cases compared to BD2DECIDE). We hypothesize that this has negatively impacted the model performance.

Another limitation is the omission of valuable semantic imaging features, qualitative imaging features that are defined by experienced radiologists (e.g. extracapsular growth, necrosis) as well as the omission of radiomics description of the GTV2 (positive lymph nodes).

Most radiomic features are designed to be extracted from a fully enclosed 3D volume, as is often the case with the primary tumor. In contrast, the peritumoral regions are rings with

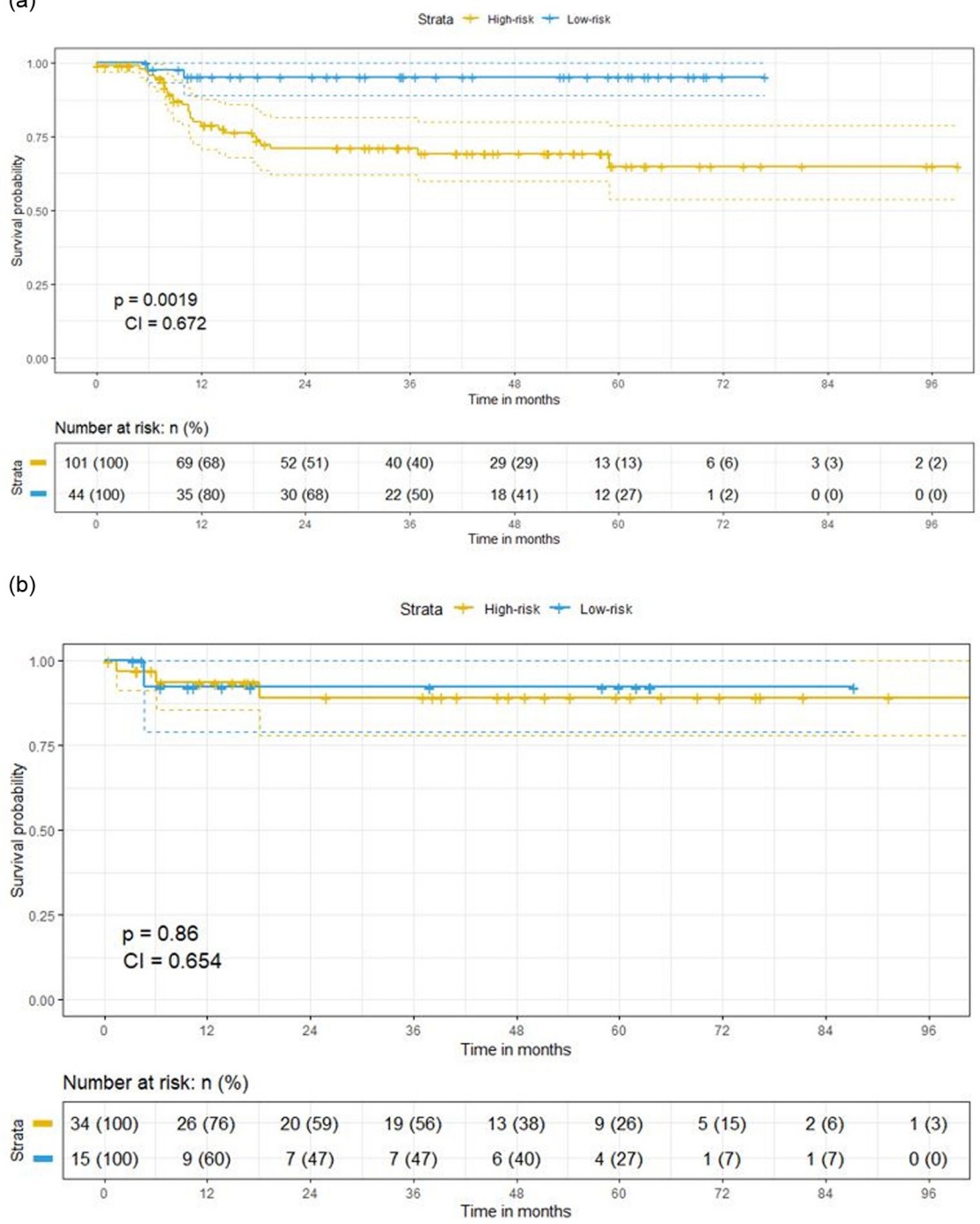

**Fig 2.** a. Training Kaplan-Meier (distant metastasis free) survival split for oropharyngeal patients (best performing clinical model in validation with Cox regression, oropharynx-DM) based on above (blue line) and below (yellow) median prediction probabilities. b. Validation Kaplan-Meier (distant metastasis free) survival split for oropharyngeal patients (best performing clinical model in validation with Cox regression, oropharynx-DM) based on above (blue line) and below (yellow) median prediction probabilities. Non-significant split in survival according to median in training, though in all of the above median cases the time to event is not observed (censoring).

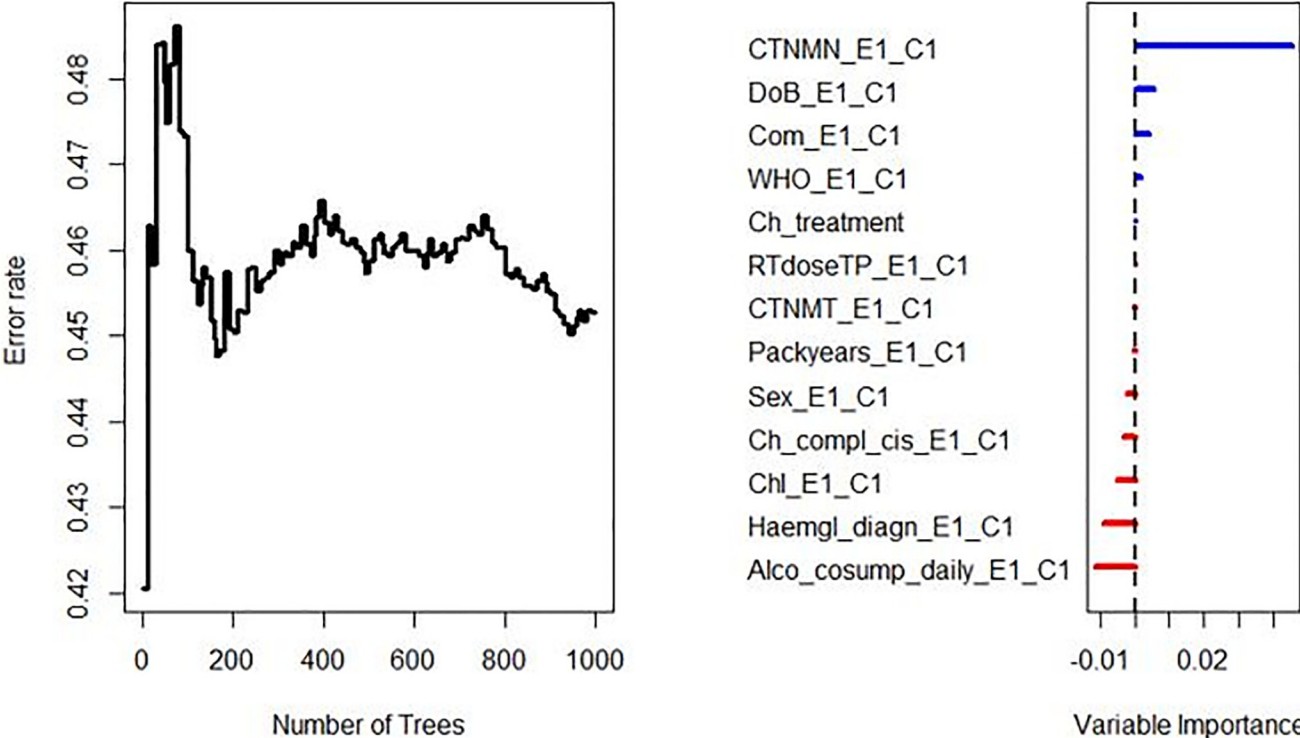

**Fig 3. Error rate stabilizes with increasing number of trees.** Features with an importance > 0 on an RFSRC model trained with all clinical variables in were eventually combined in the multivariable clinical (prognostic/ treatment-related) RFSRC model and externally validated on the BD2DECIDE dataset.

limited volume, especially the 3mm regions. Therefore, features such as those extracted from filtered images require a certain volume of the region of interest and therefore have limited application in small volumes or disjointed regions. These technical issues may have contributed to the relatively poor performance of peritumoral radiomics.

We believe that in the future, to improve clinical use of this kind of signatures, larger and more homogenous and prospectively collected data should be sought, taking into account imaging features derived from GTV2/ lymph node regions and gene expression profiles in order to construct more reliable prognostic biomarkers. An intrinsic problem might be that recurrences cannot be predicted well with bulk tumor characteristics. In a recent genetics study [23] it was shown that half of the local relapses of CRT treated HNSCCs, did not share genetic changes with the index tumors, suggesting that minor treatment resistant subclones determine outcome in many cases. Taking this into regard we believe that future radiomics studies should derive information not only from the planning CT's, but also during the multiple follow-up moments after treatment.

## Conclusion

In this study, we have investigated whether clinical data as well as computer-extracted radiomic features from peritumoral as well as inter-tumoral derived imaging features on CT can predict OS, LRR and DM. Our results show that radiomic features from the primary peritumoral regions, as well as from the primary inter-tumoral regions, do not predict OS, LRR and DM.

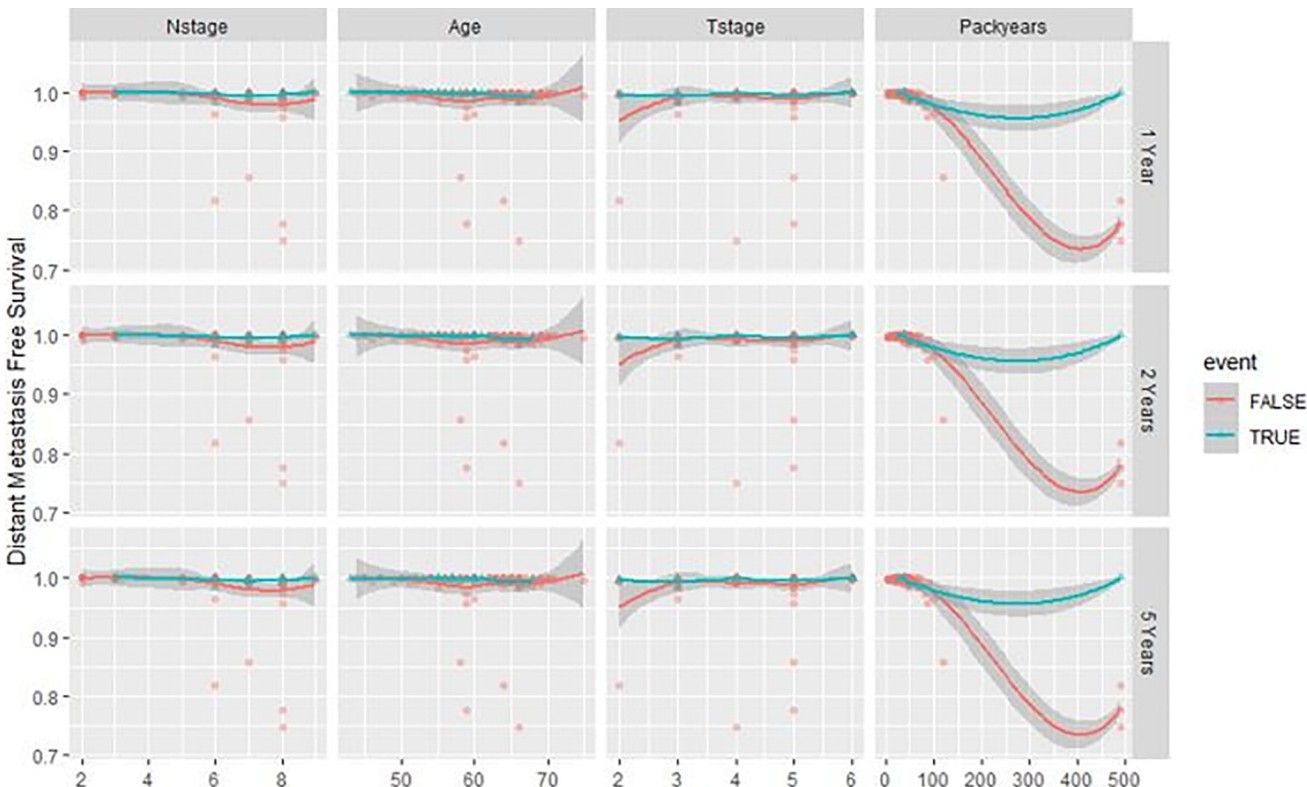

**Fig 4. Variable dependence of predicted distant metastasis at 1, 2 and 5 years on the 4 clinical variables of interest (highest performing clinical model in validation, oropharyngeal-DM) according to Random Survival Forest.** Individual cases are marked with blue triangles for censored cases and red circles for distant metastasis events. Loess smooth curve indicates the distant metastasis trend with increasing values of the individual clinical feature.

More homogenous cohorts, both in patient and imaging characteristics, and the combination of clinical, radiomics, and genomics models may increase the generalizability and predictive power of prognostic models.

## Supporting information

**S1 Appendix. Datasets, imaging parameters and missing data.**
(DOCX)

**S2 Appendix. Full names of ALL the ethics committees/institutional review boards that approved study.**
(DOCX)

**S3 Appendix. TRIPOD adherence data extraction checklist.**
(DOCX)

## Acknowledgments

Authors acknowledge financial support from ERC advanced grant (ERC-ADG-2015, n˚ 694812—Hypoximmuno), ERC-2018-PoC (n˚ 81320 –CL-IO). This research is also supported by the Dutch technology Foundation STW (grant n˚ P14-19 Radiomics STRaTegy), which is the applied science division of NWO, and the Technology Programme of the Ministry of

Economic Affairs. Authors also acknowledge financial support from SME Phase 2 (RAIL—n˚ 673780), EUROSTARS (DART, DECIDE), the European Program H2020-2015-17 (BD2Decide—PHC30-689715, PREDICT—ITN—n˚ 766276), TRANSCAN Joint Transnational Call 2016 (JTC2016 "CLEARLY"- n˚ UM 2017–8295), Interreg V-A Euregio Meuse-Rhine ("Euradiomics") and the Dutch Cancer Society.

Authors further acknowledge financial support by the Dutch Cancer Society (KWF Kankerbestrijding), Project number 12085/2018-2 and A6C 7072/2014-2.

## Author Contributions

**Conceptualization:** Simon Keek, Sebastian Sanduleanu, Frederik Wesseling, Reinout de Roest, Martijn van der Heijden, Conchita Vens, Ruud H Brakenhoff, Henry C. Woodruff, Tito Poli, Ralph Leijenaar, Frank Hoebers, Philippe Lambin.

**Data curation:** Simon Keek, Sebastian Sanduleanu, Frederik Wesseling, Reinout de Roest, Michiel van den Brekel, Lisa Licitra, Kathrin Scheckenbach, Marije Vergeer, Irene Nauta, Stefano Cavalieri.

**Formal analysis:** Simon Keek, Sebastian Sanduleanu, Reinout de Roest, Michiel van den Brekel, Martijn van der Heijden, Conchita Vens.

**Funding acquisition:** Conchita Vens, C. René Leemans, Ruud H Brakenhoff, Tito Poli, Frank Hoebers, Philippe Lambin.

**Investigation:** Frederik Wesseling, Martijn van der Heijden, Calareso Giuseppina, Kathrin Scheckenbach, C. René Leemans, Irene Nauta, Tito Poli, Philippe Lambin.

**Methodology:** Simon Keek, Sebastian Sanduleanu, Conchita Vens, Lisa Licitra, Kathrin Scheckenbach, Ruud H Brakenhoff, Irene Nauta, Henry C. Woodruff, Philippe Lambin.

**Project administration:** Reinout de Roest, Martijn van der Heijden, Calareso Giuseppina, Lisa Licitra, Kathrin Scheckenbach, Marije Vergeer, C. René Leemans, Ruud H Brakenhoff, Irene Nauta, Ralph Leijenaar, Frank Hoebers.

**Resources:** Michiel van den Brekel, Calareso Giuseppina, Marije Vergeer, C. René Leemans, Ruud H Brakenhoff, Philippe Lambin.

**Software:** Simon Keek, Sebastian Sanduleanu, Frederik Wesseling, Reinout de Roest, Henry C. Woodruff.

**Supervision:** C. René Leemans, Ruud H Brakenhoff, Stefano Cavalieri, Henry C. Woodruff.

**Validation:** Simon Keek, Sebastian Sanduleanu.

**Visualization:** Conchita Vens.

**Writing – original draft:** Simon Keek, Sebastian Sanduleanu, Frederik Wesseling, Reinout de Roest, Michiel van den Brekel, Martijn van der Heijden, Conchita Vens, Calareso Giuseppina, Lisa Licitra, Kathrin Scheckenbach, Marije Vergeer, C. René Leemans, Ruud H Brakenhoff, Irene Nauta, Henry C. Woodruff, Tito Poli, Ralph Leijenaar, Frank Hoebers, Philippe Lambin.

**Writing – review & editing:** Simon Keek, Sebastian Sanduleanu, Frederik Wesseling, Reinout de Roest, Michiel van den Brekel, Martijn van der Heijden, Conchita Vens, Calareso Giuseppina, Lisa Licitra, Kathrin Scheckenbach, Marije Vergeer, C. René Leemans, Ruud H Brakenhoff, Irene Nauta, Stefano Cavalieri, Henry C. Woodruff, Tito Poli, Ralph Leijenaar, Frank Hoebers, Philippe Lambin.

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
