## [Decision Letter · Decision Letter 0]

6 Jan 2020

PONE-D-19-31722

Computed Tomography-Derived Radiomic Signature of Head and Neck Squamous Cell Carcinoma (Peri)tumoral Tissue for the Prediction of Locoregional Recurrence and Distant Metastasis After Concurrent Chemo-radiotherapy

PLOS ONE

Dear Dr. Sanduleanu, 

Thank you for submitting your manuscript to PLOS ONE. After careful consideration, we feel that it has merit but does not fully meet PLOS ONE’s publication criteria as it currently stands. Therefore, we invite you to submit a revised version of the manuscript that addresses the points raised during the review process.

We would appreciate receiving your revised manuscript by Feb 20 2020 11:59PM. To enhance the reproducibility of your results, we recommend that if applicable you deposit your laboratory protocols in protocols.io, where a protocol can be assigned its own identifier (DOI) such that it can be cited independently in the future. For instructions see: http://journals.plos.org/plosone/s/submission-guidelines#loc-laboratory-protocols

We look forward to receiving your revised manuscript.

Kind regards,

Jason Chia-Hsun Hsieh, M.D. Ph.D

Academic Editor

PLOS ONE

Journal Requirements:

2. Thank you for your ethics statement : "According to the decisions of the Institutional Review Board and individual patient

informed consent this study was performed following the guidelines of the Code of

Conduct for Human Tissue and Medical Research (https://www.federa.org/codesconduct)

and the EU General Data Protection Regulation."

b) Please amend your current ethics statement to confirm that your named institutional review board or ethics committee specifically approved this study.

3. Please provide additional details regarding participant consent. In the ethics statement in the Methods and online submission information, please ensure that you have specified (1) whether consent was suitably informed and (2) what type you obtained (for instance, written or verbal). If your study included minors under age 18, state whether you obtained consent from parents or guardians. If the need for consent was waived by the ethics committee, please include this information.

4. We noticed you have some minor occurrence(s) of overlapping text with the following previous publication(s), which needs to be addressed:

https://doi.org/10.1371/journal.pone.0206108

In your revision ensure you cite all your sources (including your own works), and quote or rephrase any duplicated text outside the Methods section. Further consideration is dependent on these concerns being addressed.

5. Please provide an updated Competing Interests Statement in your cover letter that includes the following:

"Dr. Philippe Lambin reports, within and outside the submitted work, grants/sponsored research agreements from Varian medical, Oncoradiomics, ptTheragnostic, Health Innovation Ventures and DualTpharma. He received an advisor/presenter fee and/or reimbursement of travel costs/external grant writing fee and/or in kind manpower contribution from Oncoradiomics, BHV, Merck and Convert pharmaceuticals. Dr Lambin has shares in the company Oncoradiomics SA and Convert pharmaceuticals SA and is co-inventor of two issued patents with royalties on radiomics (PCT/NL2014/050248, PCT/NL2014/050728) licensed to Oncoradiomics and one issue patent on mtDNA (PCT/EP2014/059089) licensed to ptTheragnostic/DNAmito, three non-patentable invention (softwares) licensed to  ptTheragnostic/DNAmito, Oncoradiomics and Health Innovation Ventures. Dr. Woodruff has (minority) shares in the company Oncoradiomics. CR Leemans and RH Brakenhoff have received financial support from GenMab BV, InteRNA Technologies BV and Bristol Myers-Squibb, all outside the presented work. Ralph Leijenaar has OncoRadiomics shares."

6. We note that you have reported significance probabilities of 0 in places. Since p=0 is not strictly possible, please correct this to a more appropriate limit, eg 'p<0.0001'."

7. In your Data Availability statement, you have not specified where the minimal data set underlying the results described in your manuscript can be found. PLOS defines a study's minimal data set as the underlying data used to reach the conclusions drawn in the manuscript and any additional data required to replicate the reported study findings in their entirety. All PLOS journals require that the minimal data set be made fully available. For more information about our data policy, please see http://journals.plos.org/plosone/s/data-availability.

Additional Editor Comments:

The "clinical model" prominently featured in the results must be described in methods. Due to the negative finding, conclusions/discussion should also look into whether the finding is due to implementation or theoretical limitations.

Reviewers' comments:

Reviewer's Responses to Questions

**Comments to the Author**

1. Is the manuscript technically sound, and do the data support the conclusions?

Reviewer #1: Yes

Reviewer #2: Yes

2. Has the statistical analysis been performed appropriately and rigorously? 

Reviewer #1: Yes

Reviewer #2: Yes

3. Have the authors made all data underlying the findings in their manuscript fully available?

Reviewer #1: No

Reviewer #2: No

4. Is the manuscript presented in an intelligible fashion and written in standard English?

Reviewer #1: Yes

Reviewer #2: Yes

5. Review Comments to the Author

Reviewer #1: Major Comments:

1) In general, the paper is well written and the study is executed carefully.

2) Clinical models are a big part of results but are never defined. Please make sure to define how they are derived in methods.

3) A negative finding in an engineering paper can always mean that the method’s implementation was not good enough (e.g. different radiomics features or different preprocessing would have made the method useful). To help secure the paper’s niche in the field, please review the conclusions and limitations to focus specifically on the type of analysis and features of the data (peritumoral/inter tumoral masks) and describe why the prediction may be failing, what improvements could be made, and what is a theoretical limitation. For example, the relatively thin peritumoral layer likely poses an boundary-effect issue for wavelet decomposition, while the radiomics kernels may have a lot of missing values.

Minor Comments:

Abstract: Meaning of “clinical” model is not clear at this point in the text.

In the section “Univariable analysis and generation of multivariable models” please affirm very explicitly that the feature selection was done based on analysis of training results, not holdout data.

WHO PS is not defined

“More homogenous cohorts of patients and the combination of clinical, radiomics and

genomics models may help to generate predictive models in the future, and include

genetic/ radiomics analyses of index tumors and relapses.” – this part of conclusions maybe should be avoided. If radiomics features were found to have poor performance, their inclusion may not improve the prediction of the overall model, so this statement becomes quite speculative.

“In this study, we have investigated whether clinical data as well as computer-extracted

radiomic features from peritumoral as well as inter-tumoral derived imaging features on

CT can predict OS, LRR and DM. Our results show that radiomic features from the

primary peritumoral regions do not predict OS, LRR and DM.” – Please also state about inter-tumoral to have parallel logical flow.

Reviewer #2: This paper describes a negative finding indicating that peritumoral Radiomics from CT does not help to predict overall survival (OS), locoregional recurrence (LRR), or distant metastases (DM), which is opposite to the hypothesis. The paper is easy to follow and understand. I have the following comments that might help the authors improving the manuscript.

Based on Table 1, it seems that the validation set and training set have lots of differences (WHO, etc.), did such differences cause difficulties in the cross-validation? It seems that the authors have used a stringent validation strategy, i.e., training based on one dataset and validation using another. While it is good and desirable, case is not such ideal. Based on Table 1, it seems that the validation set and training set have lots of differences (WHO, etc.), did such differences cause difficulties in the cross-validation for the failure of replication? The 2-fold validation may also be changed to 5 fold or 10 fold if the performance is poor.

As for the multivariate Cox model, the sorted univariate-based features were gradually (forward) fed in until the first peak. This part is unclear to me. What if there is no peak or there are many trivial peaks (noisy)? How to determine a valid peak? It is helpful to understand if the authors could provide more details. Also, an occurrence frequency threshold of 50% is kind of arbitrary. Any consideration when determine such a threshold?

While multivariate Cox model is easier to understand, the multivariate version of RSF deserves a more detailed description. How does it handle the centered data for this time-to-event type of analysis? RF is easier to understand and widely used, better state the difference between RSF and RF.

Minor issues:

In the last paragraph of Introduction, “HPV-ve” should be “HPV-negative”?

“Multivariable radiomic Cox models were generated on the significant features in

univariate cox” seems redundant.

There are some typos.

“Multivariable clinical RSF models were generated based on selecting all features with a

relative feature importance >0 in the Random Survival Forest.

Multivariable radiomic RSF models were generated based on the number of features

corresponding to the first peak in C-index value in the out-of-bag cases after

consequently adding features based on decreasing relative importance in the Random

Survival Forest. The rule used to obtain the optimal amount of features is at the first

peak OR (depending on the C-index graph if there is an oscillation pattern that you

describe) after the peak until the C-index drops more than 0.02.” -- This part may contain multiple repeated descriptions and unclear descriptions. Please elaborate.

The peritumor region of interests were defined including regions 3mm and 5mm away from the tumor entity borderline. Why using two different distances? Will the results become better if increasing such a distance (e.g., to 1 cm)?

I did not see the definition of “clinical model”. Was that a model based purely on the clinical (rather than radiomics features)? Did the authors combine the clinical features and radiomics features together to predict survival? Tables 2-3 seem that the two types of methods were done separately.

6. PLOS authors have the option to publish the peer review history of their article (what does this mean?). If published, this will include your full peer review and any attached files.

Reviewer #1: No

Reviewer #2: No

---

## [Author Response · Author response to Decision Letter 0]

3 Mar 2020

Detailed response to reviewers manuscript:

‘Computed Tomography-Derived Radiomic Signature of Head and Neck Squamous Cell Carcinoma (Peri)tumoral Tissue for the Prediction of Locoregional Recurrence and Distant Metastasis After Concurrent Chemo-radiotherapy’

We would first like to thank both the editor and the reviewers for their comments to our manuscript. We have responded to each comment separately and made amendments to the manuscript accordingly. Please see our responses below. Note, each line number refers to the revised manuscript. Italic text refers to added or modified section in the manuscript.

Reviewer 1:

Major comments

1) In general, the paper is well written and study is executed carefully.

We appreciate the kind words.

2) Clinical models are a big part in results but are never defined. Please make sure to define how they are derived in the methods.

We would like to thank the reviewer for addressing this point. We have updated the methods section to include the methodology of the clinical models, similar to how the radiomics models were described. (page 8, lines 34 to 36).

“Multivariable clinical models included features selected through Cox-regression based on univariate significance (p<0.05) adjusted for multiple testing. The selected clinical features were then used to train multivariable Cox or RSF models.“

3) A negative finding in an engineering paper can always mean that the method’s implementation was not good enough (e.g. different radiomics features or different preprocessing would have made the method useful). To help secure the paper’s niche in the field, please review the conclusions and limitations to focus specifically on the type of analysis and features of the data (peritumoral/inter tumoral masks) and describe why the prediction may be failing, what improvements could be made, and what is a theoretical limitation. For example, the relatively thin peritumoral layer likely poses a boundary-effect issue for wavelet decomposition, while the radiomics kernels may have a lot of missing values.

We would like to thank the reviewer for addressing this point. We recognize we have not fully described why the peritumoral/tumoral method we have chosen fails, and a section has been added that describe these limitations, what hypothesize could improve the current method, and also why we think the current state of radiomics may also limit the possible effectiveness of the method (page 16, lines 24-29): 

“Most radiomic features are designed to be extracted from a fully enclosed 3D volume, as is often the case with the primary tumor. In contrast, the peritumoral regions are rings with limited volume, especially the 3mm regions. Therefore, features such as those extracted from filtered images require a certain volume of the region of interest and therefore have limited application in small volumes or disjointed regions. These technical issues may have contributed to the relatively poor performance of peritumoral radiomics.”

Minor comments:

 Abstract: Meaning of “clinical” model is not clear at this point of the text.

We would like to thank the reviewers for pointing out this unclarity. We have added the missing segment describing the clinical features to the abstract (page 3, methods section, line 16)

“Clinical patient features such as age, gender, performance status, etc. were collected.”

 “More homogenous cohorts of patients and the combination of clinical, radiomics and genomics models may help to generate predictive models in the future, and include genetic/ radiomics analyses of index tumors and relapses.” – this part of conclusions maybe should be avoided. If radiomics features were found to have poor performance, inclusion may not improve the prediction of the overall model, so this statement becomes quite speculative.

We would like to thank the reviewer for this comment. We have removed the relevant section from the conclusion: “and include genetic/ radiomics analyses of index tumors and relapses.”, as this is indeed speculative. The previous section we have decided to keep, as the inhomogeneity between the 2 cohorts was found to be a deciding factor of the poor performance of the different models. The modified text now reads (lines 11-13 page 17):

“More homogenous cohorts, both in patient and imaging characteristics, and the combination of clinical, radiomics, and genomics models may increase the generalizability and predictive power of prognostic models.”

“In this study, we have investigated whether clinical data as well as computer-extracted

radiomic features from peritumoral as well as inter-tumoral derived imaging features on CT can predict OS, LRR and DM. Our results show that radiomic features from the primary peritumoral regions do not predict OS, LRR and DM.” – Please also state about inter-tumoral to have parallel logical flow.

We would like to thank the reviewer for this comment. We have adjusted the section to also state that radiomics features form the primary inter-tumoral region do not predict OS, LRR and DM. Page 17,lines 8-10, now reads:

“Our results show that radiomic features from the primary peritumoral regions, as well as from the primary inter-tumoral regions, do not predict OS, LRR, and DM.”

Reviewer #2: This paper describes a negative finding indicating that peritumoral Radiomics from CT does not help to predict overall survival (OS), locoregional recurrence (LRR), or distant metastases (DM), which is opposite to the hypothesis. The paper is easy to follow and understand. I have the following comments that might help the authors improving the manuscript.

We would like to thank the reviewer for addressing these valid points that can help improve the quality of the manuscript.

Based on Table 1, it seems that the validation set and training set have lots of differences (WHO, etc.), did such differences cause difficulties in the cross-validation? It seems that the authors have used a stringent validation strategy, i.e., training based on one dataset and validation using another. While it is good and desirable, case is not such ideal. Based on Table 1, it seems that the validation set and training set have lots of differences (WHO, etc.), did such differences cause difficulties in the cross-validation for the failure of replication? The 2-fold validation may also be changed to 5 fold or 10 fold if the performance is poor.

There are indeed lots of differences between the training and validation set with regard to not only WHO PS, but also e.g. chemotherapy regimen (training mostly cisplatin alone, validation cisplatin + others and cetuximab), and clinical node stage. One of the methods that might lower the bias towards estimating the generalization errors of the clinical models would indeed be to increase the number of folds. In this case we did not increase the number of folds because we have 1) a relatively large training dataset 2) a separate validation dataset.

As for the multivariate Cox model, the sorted univariate-based features were gradually (forward) fed in until the first peak. This part is unclear to me. What if there is no peak or there are many trivial peaks (noisy)? How to determine a valid peak? It is helpful to understand if the authors could provide more details. Also, an occurrence frequency threshold of 50% is kind of arbitrary. Any consideration when determine such a threshold?

We would like to thank the reviewer for this valid concern. We have determined a valid peak according to visual inspection and if a noisy pattern existed, we determined the optimal number of features according to the point on the C-index graph after the first peak where the C-index drops no more than 0.02. The occurrence frequency threshold of 50% is indeed somewhat arbitrary, but it was simply chosen to select the features that were chosen in the majority of the models. We have added this to the methods section (page 8, line 27-33)

“These features were then ranked according to adjusted hazard ratios, where hazard ratios lower than 1 were inversed, and were gradually added to a multivariate cox model until the first peak in the cross-validation testing C-index or after the first peak until the C-index drops by more than 0.02, depending if there is an oscillation or noise pattern leading to multiple peaks. The number of occurrences of each feature in all repetitions was determined, and a selection rate > 50% was used as threshold for the final set of features, ensuring that the selected features were chosen in the majority of the models.”

While multivariate Cox model is easier to understand, the multivariate version of RSF deserves a more detailed description. How does it handle the centered data for this time-to-event type of analysis? RF is easier to understand and widely used, better state the difference between RSF and RF.

We have added more information about RSF and how this method handles a time-to-event type of analysis on page 9 line 1-6 in the methods section:

“RSF strictly adheres to the prescription laid out by Breiman (2003) and requires taking into account the outcome (splitting criterion used in growing a tree must explicitly involve survival time and censoring information) in growing a random forest model. Further, the predicted value for a terminal node in a tree, the resulting ensemble predicted value from the forest, and the measure of prediction accuracy must all properly incorporate survival information.”

Minor issues:

In the last paragraph of Introduction, “HPV-ve” should be “HPV-negative”?

Changes have been made.

“Multivariable radiomic Cox models were generated on the significant features in

univariate cox” seems redundant.

We have updated the sentence to read: ” Multivariable radiomic Cox models were generated using the significant features selected through univariate cox modelling on the training dataset.”

There are some typos.

We attempted to rectify all of them.

“Multivariable clinical RSF models were generated based on selecting all features with a

relative feature importance >0 in the Random Survival Forest.

Multivariable radiomic RSF models were generated based on the number of features

corresponding to the first peak in C-index value in the out-of-bag cases after

consequently adding features based on decreasing relative importance in the Random

Survival Forest. The rule used to obtain the optimal amount of features is at the first

peak OR (depending on the C-index graph if there is an oscillation pattern that you

describe) after the peak until the C-index drops more than 0.02.” -- This part may contain multiple repeated descriptions and unclear descriptions. Please elaborate.

Indeed multiple repeated/ unclear descriptions are given here. We have modified this sentence into:

“Multivariable radiomic RSF models were generated based on the optimal number of features corresponding to the first peak in C-index value in the out-of-bag cases or after the first peak until the C-index drops by more than 0.02, depending if there is an oscillation or noise pattern leading to multiple peaks. Hereby features with decreasing relative importance in the Random Survival Forest were consecutively added.”

The peritumor region of interests were defined including regions 3mm and 5mm away from the tumor entity borderline. Why using two different distances? Will the results become better if increasing such a distance (e.g., to 1 cm)?

Ideally, we would have liked to do a sensitivity analysis in which we gradually increase the peritumoral regions of interest. Nevertheless, too small regions would have resulted in less radiomics features being calculated (some filters cannot be applied to small regions of interest) and too large peritumoral sizes would have resulted in many regions falling outside of the anatomical boundaries of the H&N region (e.g. in bone, cartilage and air).

I did not see the definition of “clinical model”. Was that a model based purely on the clinical (rather than radiomics features)? Did the authors combine the clinical features and radiomics features together to predict survival? Tables 2-3 seem that the two types of methods were done separately.

The clinical model was indeed based solely on clinical variables. It seems that both clinical as well as radiomics features are performing poorly in predicting any outcome. Combining these two methods would not rely in any major improvement and would most likely confuse the readers. We have therefore chosen to omit this additional analysis.

To clarify the clinical model building we have updated the text to read (page 8 lines 34-38):

“Multivariable clinical models included features selected through Cox-regression based on univariate significance (p<0.05) adjusted for multiple testing. The selected clinical features were then used to train multivariable Cox or RSF models.

Multivariable clinical RSF models were generated based on selecting all features with a relative feature importance >0 in the Random Survival Forest.”

---

## [Decision Letter · Decision Letter 1]

20 Apr 2020

Computed Tomography-Derived Radiomic Signature of Head and Neck Squamous Cell Carcinoma (Peri)tumoral Tissue for the Prediction of Locoregional Recurrence and Distant Metastasis After Concurrent Chemo-radiotherapy

PONE-D-19-31722R1

Dear Dr. Sanduleanu,

We are pleased to inform you that your manuscript has been judged scientifically suitable for publication and will be formally accepted for publication once it complies with all outstanding technical requirements.

With kind regards,

Jason Chia-Hsun Hsieh, M.D. Ph.D

Academic Editor

PLOS ONE

Additional Editor Comments (optional):

All the questions from the reviewers were answered adequately.

Reviewers' comments:

Reviewer's Responses to Questions

**Comments to the Author**

1. If the authors have adequately addressed your comments raised in a previous round of review and you feel that this manuscript is now acceptable for publication, you may indicate that here to bypass the “Comments to the Author” section, enter your conflict of interest statement in the “Confidential to Editor” section, and submit your "Accept" recommendation.

Reviewer #2: All comments have been addressed

2. Is the manuscript technically sound, and do the data support the conclusions?

Reviewer #2: Yes

3. Has the statistical analysis been performed appropriately and rigorously? 

Reviewer #2: Yes

4. Have the authors made all data underlying the findings in their manuscript fully available?

Reviewer #2: No

5. Is the manuscript presented in an intelligible fashion and written in standard English?

Reviewer #2: Yes

6. Review Comments to the Author

Reviewer #2: (No Response)

7. PLOS authors have the option to publish the peer review history of their article (what does this mean?). If published, this will include your full peer review and any attached files.

Reviewer #2: No

---

## [Editor Report · Acceptance letter]

7 May 2020

PONE-D-19-31722R1 

Computed Tomography-Derived Radiomic Signature of Head and Neck Squamous Cell Carcinoma (Peri)tumoral Tissue for the Prediction of Locoregional Recurrence and Distant Metastasis After Concurrent Chemo-radiotherapy 

Dear Dr. Sanduleanu:

I am pleased to inform you that your manuscript has been deemed suitable for publication in PLOS ONE. Congratulations! Your manuscript is now with our production department. 

With kind regards,

on behalf of

Dr. Jason Chia-Hsun Hsieh 

Academic Editor

PLOS ONE